# Application of Dynamic Analysis Methods into Assessment of Geometric Properties of Chalcedonite Aggregates Obtained by Means of Gravitational Upgrading Operations

**Tomasz Gawenda, Damian Krawczykowski, Aldona Krawczykowska** , **Agnieszka Saramak \*** **and Alona Nad**

Department of Environmental Engineering and Mineral Processing, AGH University of Science and Technology, 30-059 Cracow, Poland; gawenda@agh.edu.pl (T.G.); dkrawcz@agh.edu.pl (D.K.); aldona.krawczykowska@agh.edu.pl (A.K.); alonanad@agh.edu.pl (A.N.)
\* Correspondence: saramak@agh.edu.pl

**Abstract:** The aim of the paper is an assessment of geometrical properties of regular and irregular particles of chalcedonite enrichment products carried out in a laboratory ring jig. The investigative program included experiments of aggregate enrichment, along with visual analyses made for the obtained products, separately for regular and irregular particles. Several shape coefficients were calculated, and the most effective ones in terms of assessment of particle regularity were selected from among them. Particle size distributions for feed and enrichment products were also determined using the idea of minimum Feret's diameter, and the intensity of dust emission by individual products was measured as well. The results obtained by the visual system were discussed in the context of their application in the assessment of enrichment operations carried out in a water jig.

**Keywords:** particle shape analysis; aggregate enrichment; jig; chalcedonite

---

## 1. Introduction

Shape of particles, especially in terms of their regularity and irregularity, to a large extent affects the quality of mineral aggregates. The yield of regular or irregular particles in aggregate products, in turn, strictly depends on the technology of feed material crushing and its geomechanical properties. The highest quality aggregate products should contain mostly cubic particles, while the lower quality fractions contain flat particles, with a large surface-to-volume ratio. There have been many studies carried out on the evaluation of the properties of regular and irregular particles and their impact on the quality of product aggregates, in particular concretes and asphalt mixtures [1–7]. In the geomechanics of soils, the shape of particles affects their strength, stability, degree of compaction and other mechanical parameters [8–11]. The particle shape also plays an important role in the technologies of mechanical processing of raw materials, especially in hydraulic classification, gravity beneficiation, and compaction [12–14]. The regularity of particles also has a significant impact in concrete production [15] as well as in dust generation [16].

The particle shape is a feature that can be regarded from different scopes, determined according to different classifications, and by using different shape factors and measurement techniques. For example, the Zing classification [17,18] groups the particles into cubic, flat, plane–column and column. In industrial production of road and building aggregates, two standards: EN 933-3: 2012 [19] and EN 933-4: 2008 [20] are used for determining geometric properties of product particles in terms of shape (FI) and flatness (SI) indices [21].

---

Some more advanced methods of quantitative image analysis based, for example, on Fourier descriptors [22,23] are known in the literature. However, these methods are useful only in simple research because the degree of complexity limits their practical application to the particle shape characteristics. In order to change this approach, a simpler methodology identifying the particles' morphology should be proposed, based on the analysis of their image, which would generate only relevant and easily interpretable descriptive parameters, for example: convexity, circularity or aspect ratio.

Modern video techniques, utilizing computer image analysis for determining the shape of particles, are becoming more commonly applied in particle shape evaluation. Even if technological development heading towards automation of measurements (i.e., microscopic pictures) allows for increasing the representative number of particles in the measurement, the methods based on the microscope generally include 2D image analysis. As a result, to some extent an incomplete information is generated because the analyzed particles are always situated on a plane in their most stable position [24,25]. Three-dimensional (3D) geometric data can be obtained by means of advanced optical microscopy, i.e., Axiovision's topographic module [26] or computed tomography [27,28], however the complexity of these methods limits their practical application only to basic research. Figure 1 shows the idea of spatial dimensioning of a single particle. The possibility of obtaining a large number of particle images in a short period of time, in a way that the results are statistically representative, is also a problem here. It seems that video analyzers based on the dynamic image analysis method (DIA) are the most suitable devices that might help in overcoming this problem.

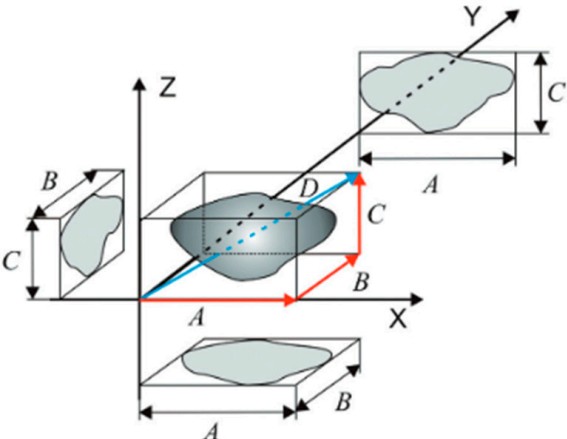

**Figure 1.** A scheme of determination of basic dimensions of a particle. Red arrows denote projections of main dimensions, blue arrow—maximum diagonal of particle [29].

The article evaluates an application of the dynamic analysis technique for the assessment of a particle's shape by means of commonly known indices: aspect ratio, convexity and circularity. The particle size distribution of regular and irregular particles was also determined, utilizing the minimum Feret's diameter as a measure. A novelty in the approach concerns adoption of this technique into qualitative and quantitative description of jig separation products. The results of investigation showed that by means of visual systems it can be possible to have a distinction in shape regularities/irregularities, especially in enrichment of raw feed mixture containing both regular and irregular particles [30]. In such an approach, this system can be a supplementary or alternative method in the description of jigging process effects, compared to standard method used [31,32]. The dynamic image analysis technique is also considered to be a credible method for the qualitative assessment of jig enrichment process results.

## 2. Materials and Methods

The measuring visual system Analysette 28 ImageSizer (produced by Fritsch, Idar-Oberstein, Germany), utilizing a dynamic image analysis, was used in investigations. The analyzer generates a stroboscopic effect from a pulsating light source, while the camera and interchangeable optical system record images at high speed. The visual system records the images of particles in motion, with the speed of up to several hundred objects per second. The measurement is carried out for dry particles, falling down in front of the camera lens. The registered particles are then subjected to computer image analysis, on the basis of which their basic geometrical parameters are calculated. The principle of operation of the visual system is similar to the measurement using a microscope—the camera records an image of particles, and then the computer software analyzes the shape and size of each particle separately. Microscopic measurement, however, is a static measurement and allows for analyzing only a few particles at a time. The DIA method makes it possible to measure particles in motion. Free movement leads to a random orientation of particles and actual shape and size of individual particles can be determined very accurately. The continuous feeding of dispersed particles results in reliable and representative measurement results, obtained on the basis of a large number of registered particles at a high level of statistical confidence. Thanks to such a method of measurement, approximately ten thousand images per minute can be analyzed, not just a few. The concept of the measurement is shown in Figure 2.

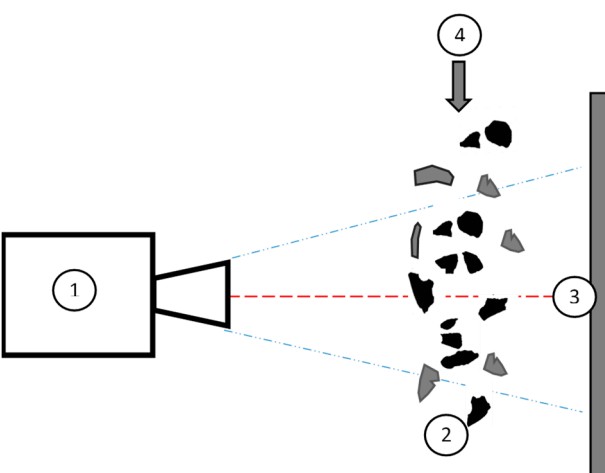

**Figure 2.** The scheme of dynamic image analysis (DIA) measurement system: 1—camera, 2—measurement volume, 3—light source, 4—feeding direction of particles.

The original results obtained by means of visual methods (an image analysis) are based on numerical distributions. They are often converted into volume distributions and this is an acceptable procedure, because image analysis provides much more information than any other particle size measurement technique. The measurement of each particle provides the user with great possibilities of calculating and reporting particle sizes, e.g., image analysis systems can generate distributions based on particle length or volumetric distributions that are based on shapes other than spheres. These possibilities are described in the ISO 13322-2 standard [33], developed for the dynamic image analysis (DIA) method. For the description of particle size distribution, single values (an average, for example) should not be used. A good practice for this method is to provide diameters D10, D50 and D90 [34], as well as full distributions and the value of Span coefficient [35].

## 3. Investigative Program

The material used in investigations was a chalcedonite within particles with a size fraction 6.3–8 mm, previously separated from a bulk aggregate feed. Samples of regular and irregular particles were separated by means of a slotted sieve bar. For the purposes of this paper, samples containing

irregular particles were denoted as (Irr) while a sample with regular particles was described as (Reg). Scheme of investigations is presented in Figure 3. Feed and products of both samples were subjected to a visual analysis, marked in grey dots and described A1 to A5 (for regular particles) and B1 to B5 (for irregular ones).

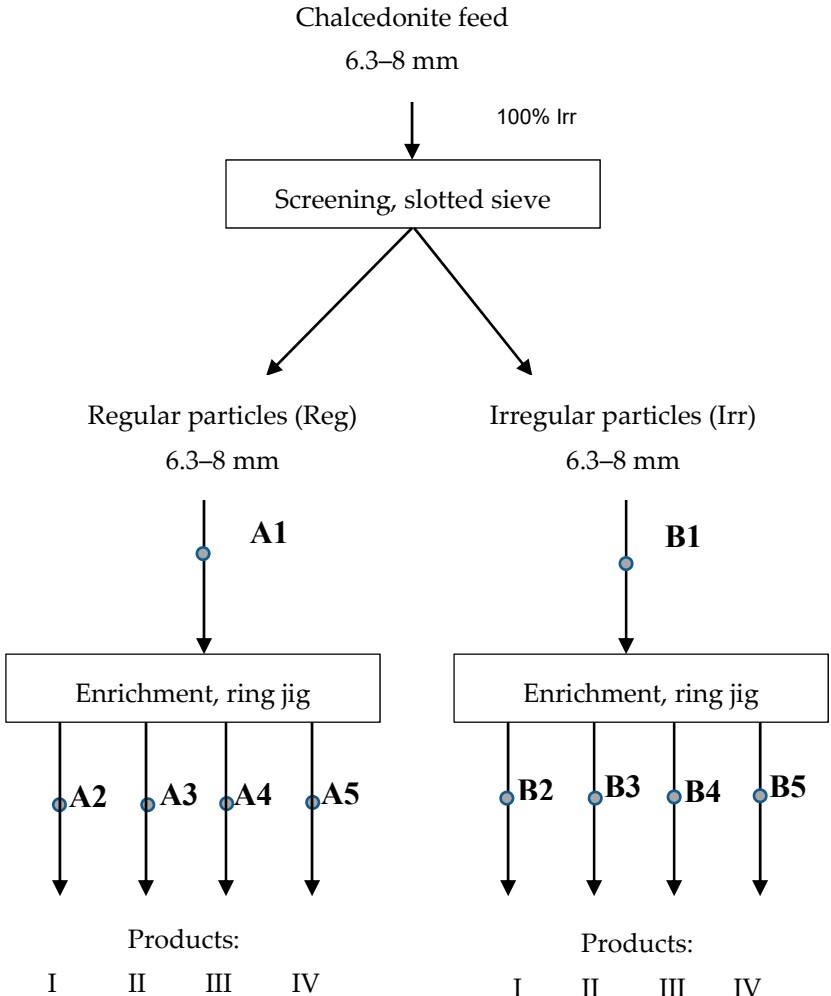

**Figure 3.** Scheme of investigations.

Preparation of samples with regular and irregular particles was performed according to the patents listed in Section 7. These innovative solutions were also tested in [36]. The material was then subjected to the enrichment process in a laboratory ring jig, and after 5 min of the device's operation, various products located in individual layers of the jig bed were obtained. The enrichment process in the jig and placement of individual products are presented in Figure 4. The laboratory ring jig has a working chamber with a diameter of 200 mm and the maximum height of 600 mm. The water can pulsate up to 120 cycles per minute. It is possible to obtain up to 16 layers of enriched products on various heights of the working chamber of the jig.

As can be seen on Figures 3 and 4, in the jig enrichment process a total of four products, one from each layer of the jig bed, was obtained. The first product (I) was the bottom one, while the fourth (IV) was the upper layer of the material (Figure 3). In addition to the feed, the products of regular and irregular particles enrichment were also subjected to analyses aimed at determining differences in the geometrical parameters of particles, and thus, at examining the effect of aggregate beneficiation. It is therefore one of the paper's aims and the main reason of jig process application to show a shape differentiation in the particles in individual jig layers. According to the adopted

methodology, the raw feed was divided into samples containing regular (Reg) and irregular (Irr) particles. Their shape was characterized according to the standard No. PN-EN 933-3: 2012 [37], part III, used in aggregate production.

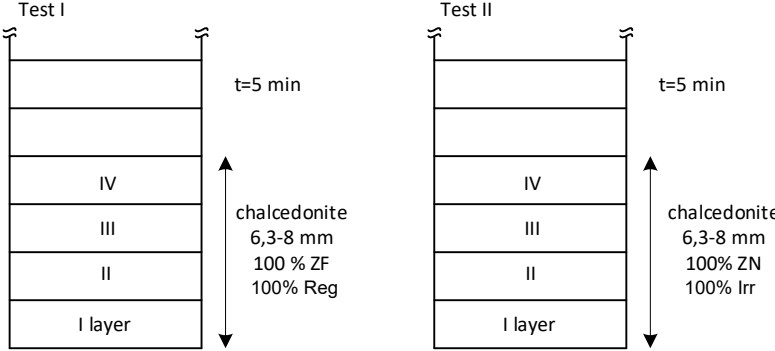

**Figure 4.** Scheme of investigations for testing of regular and irregular particles in particle size fraction 6.3–8 mm in laboratory ring jig device.

The following characteristics on jig products, separately for regular (Reg) and irregular (Irr) particles, were specified:

(1) Particle size distributions on the basis of the minimal Feret's diameter (dFmin). Graphical interpretation of this diameter is shown in Figure 5a. It is a minimal diameter of a particle which is placed on a plane in its most stable position.

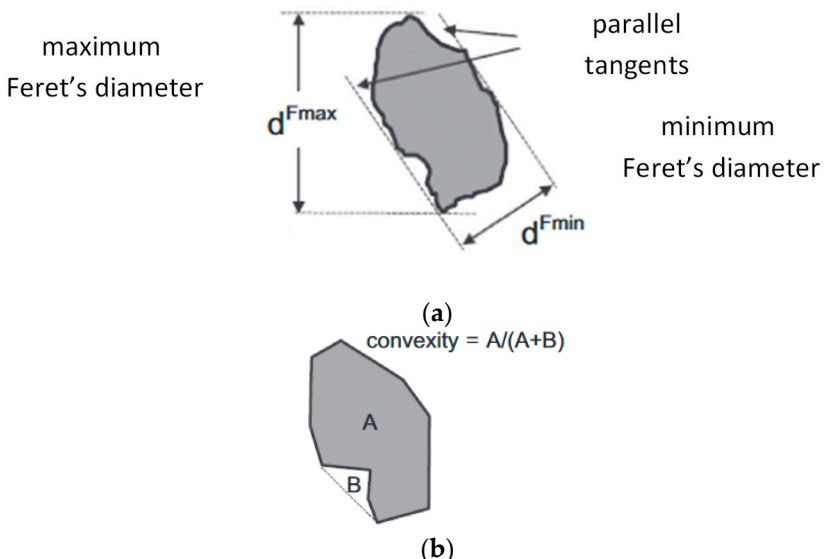

**Figure 5.** (**a**) Idea of Ferets' diameters; dFmin—minimal Feret's diameter, dFmax—maximum Feret's diameter [38]. (**b**) Particle convexity; A—convex surface of the particle, B—concave surface of the particle [38].

(2) *AR* (aspect ratio), defined as the proportional relationship between the particle width and height, and determined from formula:

$$AR = \frac{d_{Fmin}}{d_{Fmax}}$$

(3) *Cx* (convexity), the relation of perimeter length of the convex hull (envelope) that bounding the particle to the length of particle perimeter (Figure 5b):

$$Cx = \frac{A}{(A + B)}$$

(4) *C* (circularity), defined as the relationship of a particle's total surface and the Crofton parameter, according to formula:

$$C = \frac{4\pi D}{P_c{}^2}$$

where: *D* is the particle's surface and $P_c$ is the normalized measure of the number of edges of 0°, 45°, 90° and 135° in the object (Crofton parameter).

(5) Determination of dust emission for selected jig products and the feed. Total suspended particulates (TSP, [mg/m³]) was determined, which in real conditions means the suspension of particles with a diameter finer than 20 m. Measurements were performed using a Casella Microdust Pro meter (Figure 6), and the measurement interval was one second.

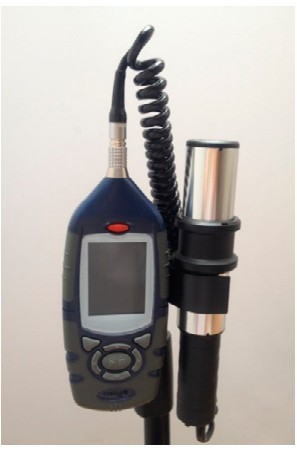

**Figure 6.** Casella measuring device.

## 4. Results

*4.1. Quantitative Characteristics of Particles*

Figures 7 and 8 show shapes of regular and irregular chalcedonite particles generated by means of computerized image analysis. As it was mentioned earlier, the shape was characterized according to the standard PN-EN 933-3: 2012 [37]. The norm defines the regular particles as those with the length not exceeding three times the width and thickness. The reason that DIA analysis was used separately in shape determination for regular and irregular particles is because application of various visual methods can be helpful in shape analysis for products of the plant scale jigging process, as well as regular aggregate production and enrichment according to the patented solution. These methods can replace mechanical methods used in particles shape control [32].

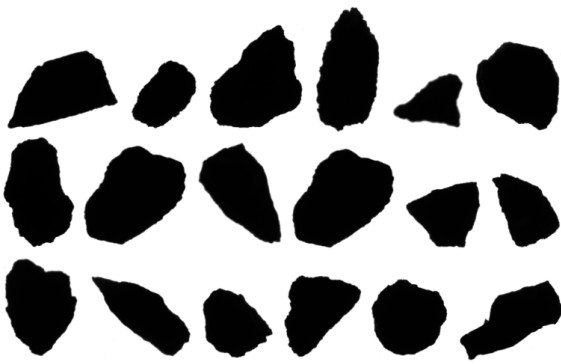

**Figure 7.** Regular particles of chalcedonite feed in particle size fraction 6.3–8 mm.

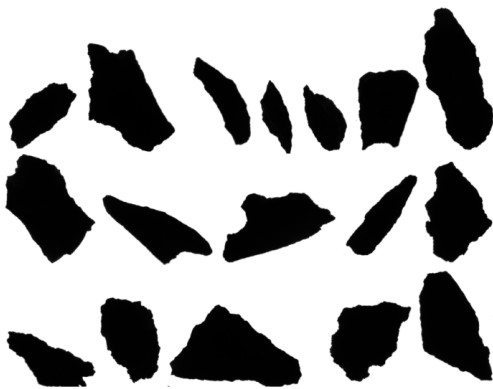

**Figure 8.** Irregular particles of chalcedonite feed in particle size fraction 6.3–8 mm.

An analysis indicates the visible sharpness and irregularity in shapes of irregular particles compared to regular ones. Extreme lengthening of size proportions for some irregular particles in relation to regular ones in 2D projection, as well as their irregular outline, were obtained for the same particle size class but separately for regular and irregular particles, indicating a much higher flatness of irregular particles.

Quantitative geometry characteristics of selected regular and irregular particles are shown in Figures 9 and 10. Many parameters and shape coefficients have been presented here, describing in detail the geometrical properties of each registered particle, such as: area, circumference (perimeter), equivalent diameters (equivalent diameter), Feret's diameter, ellipsis fit, or the coefficients of solid and particle shape (morphology), as well as the image focus parameter. Depending on the needs, individual parameters give an opportunity for detailed analysis of particle geometry. Such characteristics were performed for each single particle within the sample, which served as the base for further characteristics in terms of particle size, convexity, circularity and aspect ratio indices.

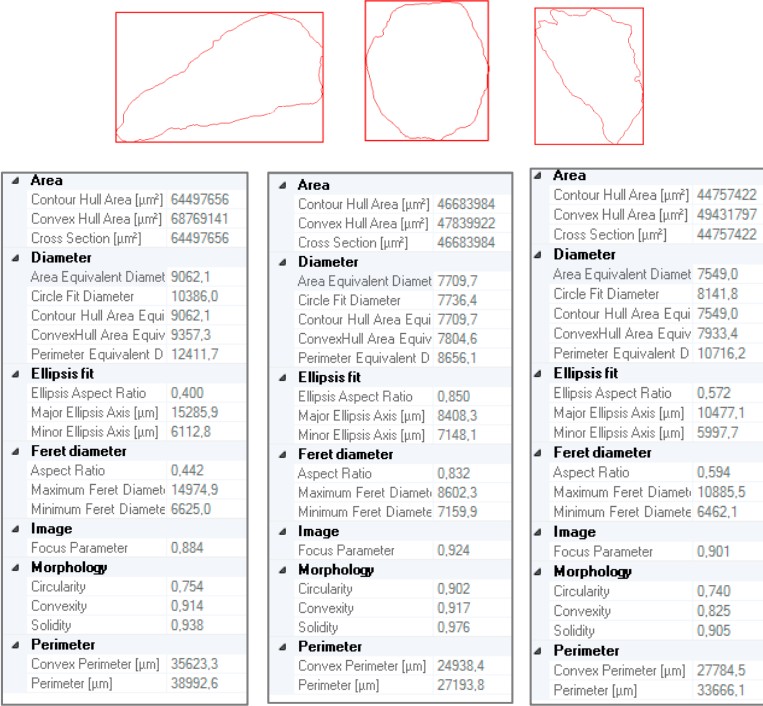

**Figure 9.** List of determined geometrical parameters for selected regular particles of chalcedonite.

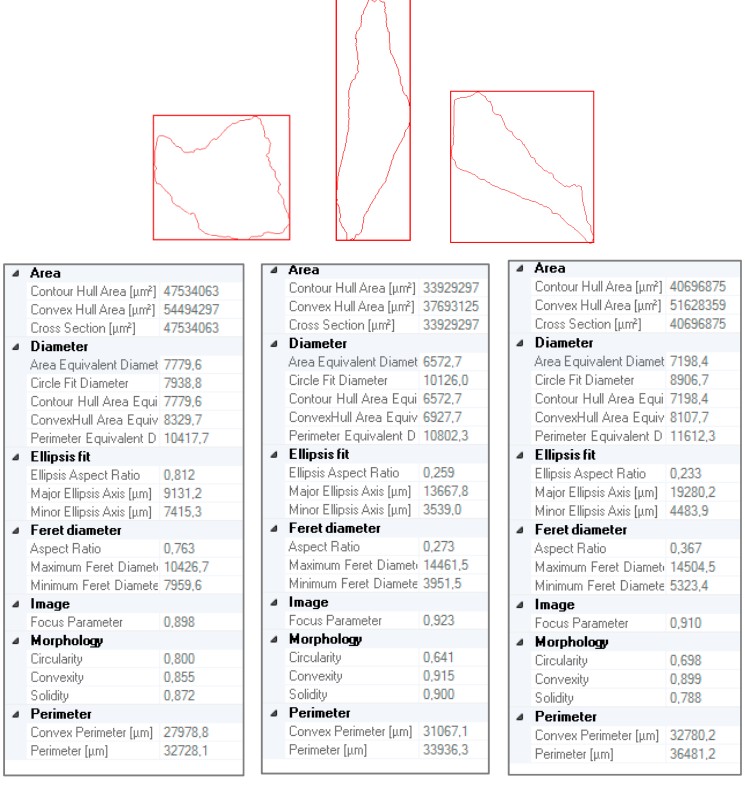

**Figure 10.** List of determined geometrical parameters for selected irregular particles of chalcedonite.

## 4.2. Determination of Shape Coefficients

The next stage of analysis included determination of selected shape coefficients (*AR*, *C*, *Cx*) along with their distributions, separately for regular and irregular particles. Averaged results are presented in Figure 11.

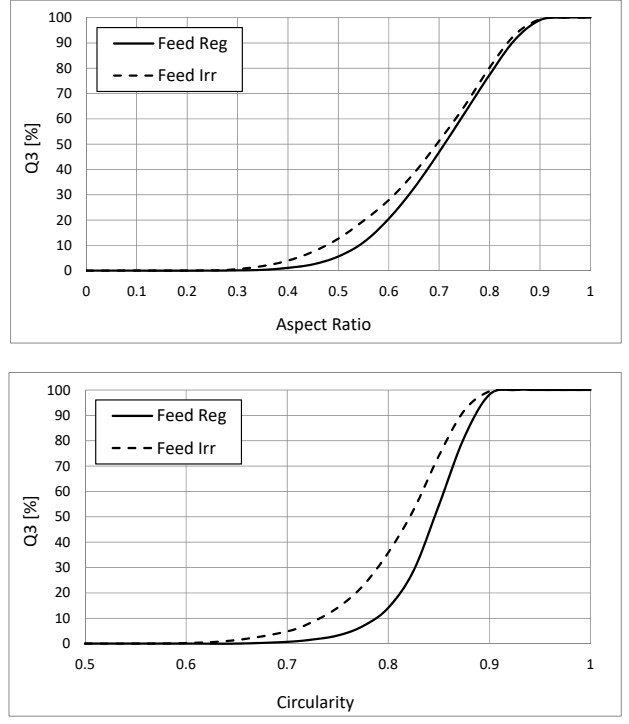

**Figure 11.** *Cont.*

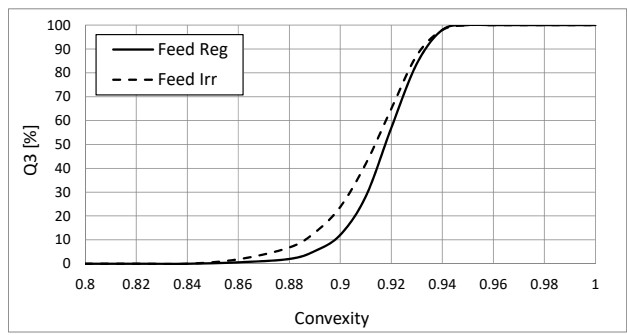

**Figure 11.** Distribution of selected shape coefficients for regular and irregular particles in feed material.

Analysis of these graphs clearly shows that the coefficient, which apparently indicates differences in chalcedonite particle size, is circularity (*C*). Average values of the calculated shape coefficients for feed with regular particles are respectively: *AR*mean = 0.708, *Cx*_mean = 0.918, *C*mean = 0.847, while for irregular ones the results were lower: *AR*mean = 0.696, *Cx*_mean = 0.914, *C*mean = 0.822. However, the two remainings coefficients also show variations for regular and irregular particles, but it is not as large as in the case of the *C* coefficient.

### 4.3. Analysis of Particle Size Distributions for Feed and Products of the Jig

Variations in shape coefficient values obtained for regular and irregular particles could indicate for various particle size compositions of both samples. The performed analysis of the particle size distribution according to the minimum Feret's diameter dFmin carried out for the tested feed with regular and irregular particles, prepared on slotted sieves, indeed showed the differences in their particle size (Figure 12). The feed material with irregular particles contained twice as many fine particles (below 5 mm). The visual system for these particles dynamically recorded the smallest third dimension of irregular particles, indicating their flatness, as the minimum Feret's diameter and thus decreased the result of particle distribution. In the area of intermediate particles physically measured on 6.3 mm and 8 mm slot sieves, dynamic image analysis measuring their particle size distribution according to dFmin underestimates the proportion of irregular particles in the feed. However, it overestimates the yield of these particles in the upper particle size range, close to the size of 8 mm, obtained on a slotted sieve. This is due to the fact that irregular particles are randomly registered by the visual system, as well as in the orientation displaying the two largest dimensions, without capturing the third—the smallest, and that one gives information regarding the flatness (irregularities) of these particles. These two larger dimensions of irregular particles are usually greater than the corresponding dimensions of regular particles within the same particle size class. Differences in the distribution of particles between regular and irregular ones are therefore directly related to differences in their shape. It can be assumed that the differences would be greater with the use of the 3D technique, which in each case takes into account the third dimension of the particle, and not only randomly in the case of a convenient orientation of the particle, such as in the case of the DIA method.

The products of chalcedonite enrichment, containing regular and irregular particles, were subjected to further visual analyses. Figure 13 presents the particle size distributions for regular and irregular particles of chalcedonite, obtained for individual enrichment products in the jig along with the feed. The results indicate that the feed containing regular particles and its enrichment products (I, II, III, IV) obtained in different zones (heights) of the jigging beds have almost identical particle size compositions. A similar comparative analysis for irregular particles indicates small differences in the particle size of individual enrichment products, but it also shows that the bottom product (I) especially has significantly finer graining in relation to the feed. Fine particles in enrichment products of irregular particles could be generated due to mechanical breaking of flat particles, which are also weaker. Confirmation of this phenomenon can also be seen in the variations of the circularity and convexity shape coefficients, especially in terms of the feed material in relation to the enrichment products from the lowest layer

(I) (Figure 14), as well as in various levels of the dust emission (see Section 4.4). The fine particles that were formed in that manner have, to a large extent, the circular shapes. This is confirmed (Figure 15) by concentration of these particles in the upper left corner of the graph (black dots), which are characterized by high values of the *C* aspect ratio within the fine particles below 4 mm. Analyzing the distribution of the convexity coefficient (Figure 14), it can be also seen that irregular particles of chalcedonite (flat, often fragile with low strength properties) after the enrichment process in the water jig have a smoother surface (higher *Cx* coefficient in relation to the feed). This may be due to the washing effect of water.

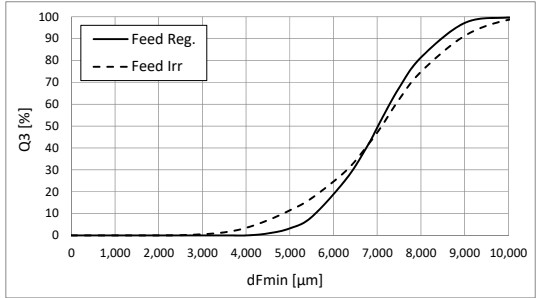

**Figure 12.** Cumulative particle size distribution functions of chalcedonite feed with regular (Reg) and irregular (Irr) particles.

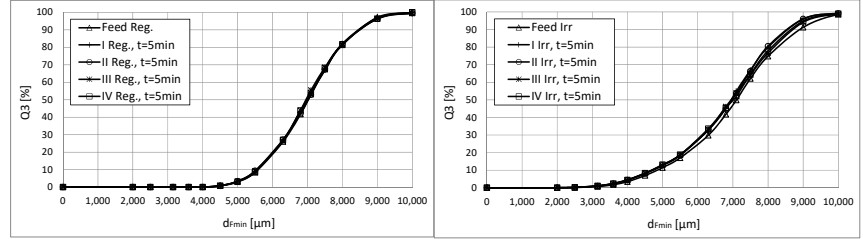

**Figure 13.** Particle size distribution curves for feed and enrichment products with regular and irregular particles.

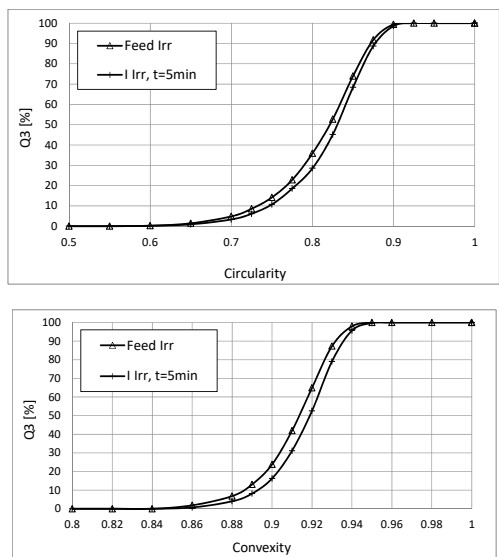

**Figure 14.** Distributions of convexity and circularity indices for feed and enrichment products of I and IV layer—separately for regular and irregular particles.

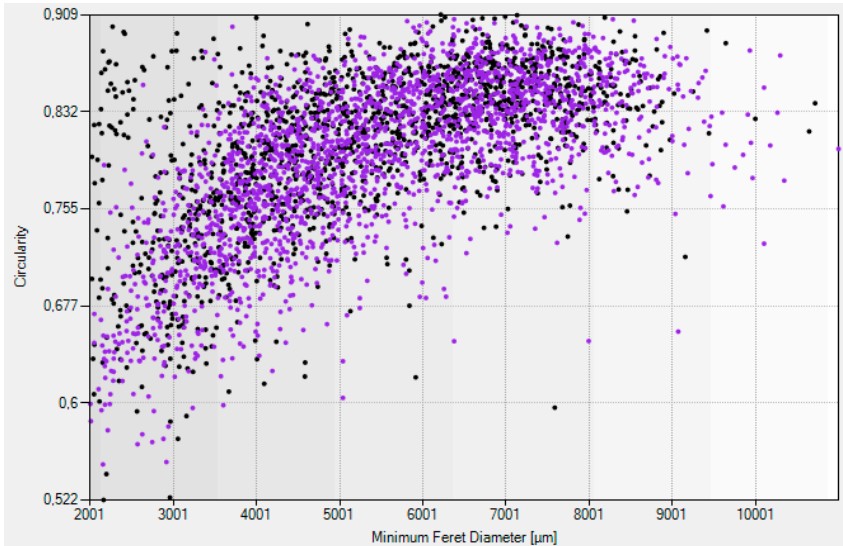

**Figure 15.** Distribution of the convexity index for irregular particle feed and its product from the bottom layer (I) in relationship to the particle size.

### 4.4. Dust Emission

Irregularities in particle size distributions, as well as various values of selected shape factor may also indicate the various content of very fine particles in individual samples. This may be effective in different enrichment performances depending on whether the particle is regular or not. Considering the shapes of particles, especially for flat shapes, elongates particles with sharp edges, in operations of material handling a more intense dust generation, resulting from micro breaking or particle corner abrasion, can take place. Measurements for both feed samples (i.e., regular and irregular particles) were performed by means of the Casella dust measuring device. Figures 16 and 17 present the registered dust emission for Reg and Irr feed along with base characteristics, while Table 1 summarizes the results obtained for all enrichment products.

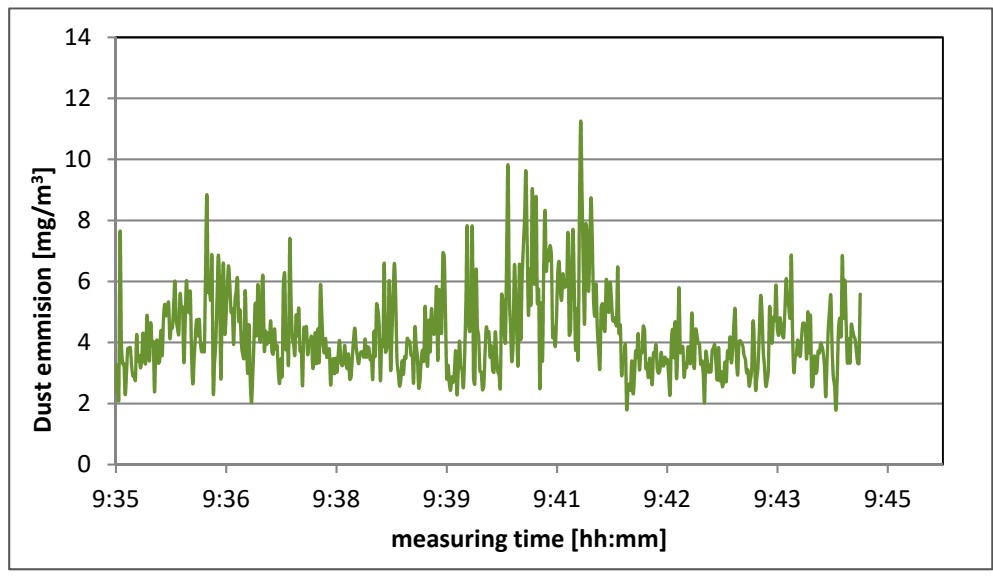

**Figure 16.** Dust generation for irregular particles—feed material $\bar{x} = 4.91$, $s = 1.84$.

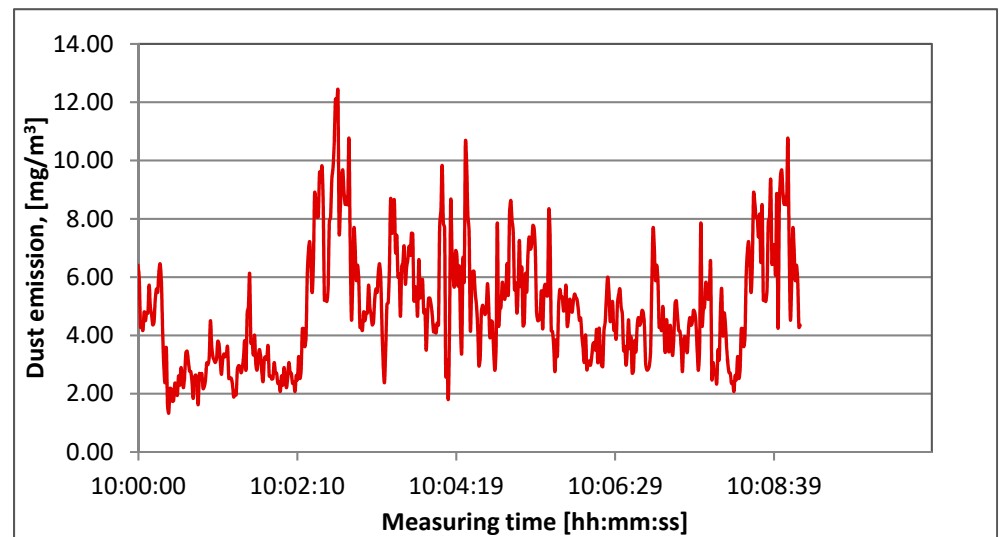

**Figure 17.** Dust generation for irregular particles—feed material $\bar{x} = 4.46$, $s = 1.30$.

**Table 1.** Values of dust generation [mg/m$^3$] for individual enrichment products.

| Parameter | Reg Enrichment Products | | | | Irr Enrichment Products | | | |
|---|---|---|---|---|---|---|---|---|
| | I | II | III | IV | I | II | III | IV |
| Average, $\bar{x}$ | 4.42 | 6.45 | 7.51 | 6.58 | 5.01 | 6.24 | 7.34 | 6.92 |
| Standard deviation, $s$ | 1.28 | 1.99 | 3.32 | 2.72 | 1.92 | 2.83 | 3.26 | 2.65 |

Variation between average values of dust emission for two types of feed is statistically significant on the accepted probability level $1 - \alpha = 0.95$. For enrichment products, in turn, statistically significant differences in average dust emission were observed for the bottom product (layer I) and the upper product obtained in zone IV—marked in the table in red, which is convergent with the results presented in Figures 16 and 17. In general, an analysis of dust emission shows that depending on the regular and irregular particles, dust emission in material handling are different.

## 5. Discussion

Comparing the shape coefficients of particles in individual layers of the jig, differences can be seen in their values between the lowest (I) and upper (IV) layer. Particles in the bottom layer have a smoother surface (higher values of $Cx$ coefficient) and a more circular shape (higher $C$ coefficient values), while particles in the upper layers have a more irregular surface and less spherical shapes (Figures 18 and 19). The speed of sinking for these particles is therefore lower, and for that reason, they concentrate in the top layers of the jig bed. This shows that the enrichment effect occurs even within the same category of regular and irregular particles. In the mixture of regular and irregular particles, the effect of separation, and thus enrichment of the mixture would be stronger.

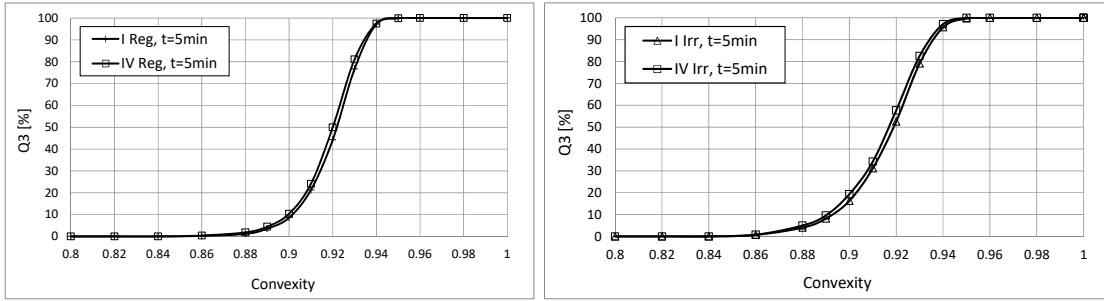

**Figure 18.** Distributions of convexity index for enrichment products I and IV for regular and irregular particles.

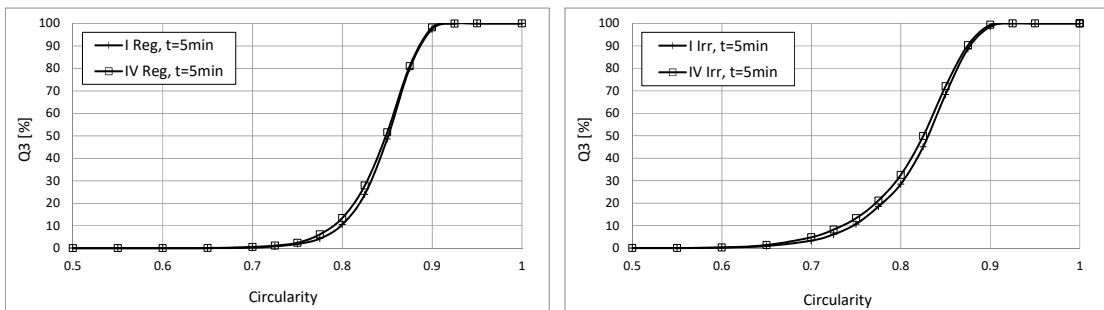

**Figure 19.** Distributions of circularity index for enrichment products I and IV for regular and irregular particles.

An analysis of summary diagrams with the distributions of the Feret's minimum diameters and convexity and circularity coefficients of both studied populations of particles (Figures 20–22) show the differences in particle size, surface structure and the shape of regular and irregular particles for feed and the products. Samples with irregular particles were characterized by a higher yield of fine grains (Figure 20), had less circular shape (Figure 21) and irregular surface (Figure 22).

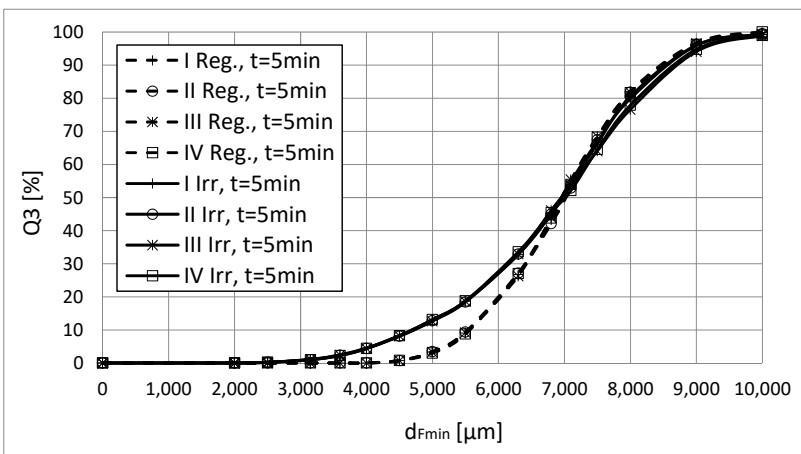

**Figure 20.** Comparison of particle size distributions of chalcedonite enrichment products for regular and irregular particles.

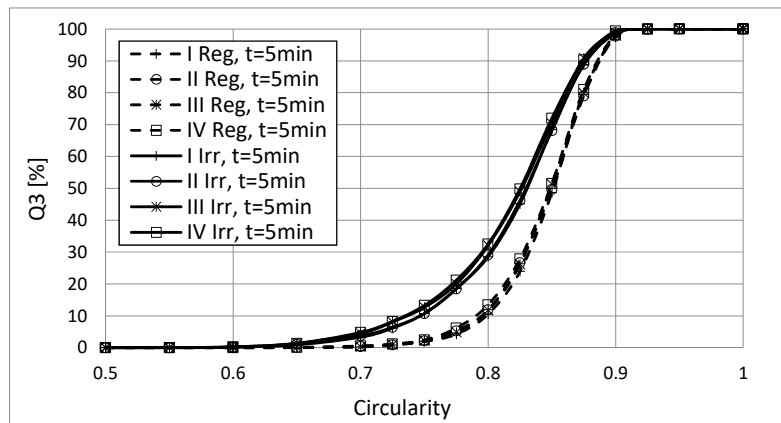

**Figure 21.** Comparison of Circularity distributions of chalcedonite enrichment products for regular and irregular particles.

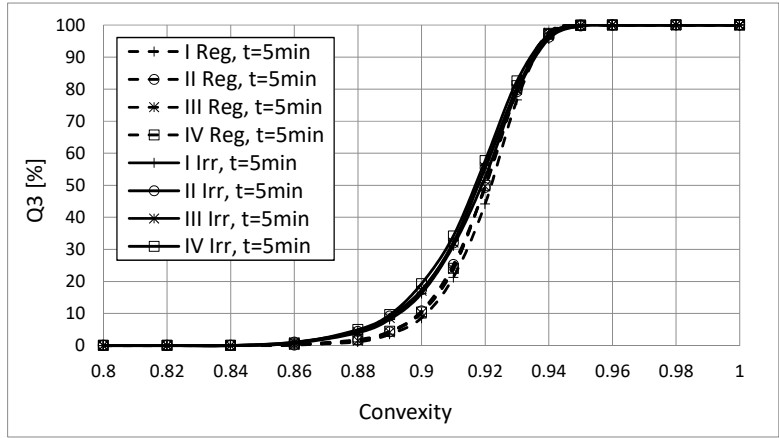

**Figure 22.** Comparison of Convexity distributions of chalcedonite enrichment products for regular and irregular particles.

## 6. Summary and Conclusions

The results of investigations show that the shape of particles and the choice of a measure for determining the size of the particle significantly affect the results of particle size measurements by means of visual methods. The dynamic image analysis (DIA) technique gives repeatable and reliable results in the size and shape of chalcedonite particles, differentiating them for regular and irregular grains. The differences in the surface structure and geometry of these particles both for the feed and enrichment products from the individual layers of the jig, can be observed. The circularity coefficients $C$ and convexity $Cx$ proved to be the effective coefficients of shape, while the minimum diameter of Feret's $Fmin$ was the effective measure of the size.

Relatively low variations in values of shape coefficients could be observed during investigations. It is due to the fact that a separate enrichment of regular and irregular particles was analyzed. However, as can be seen in Figures 20 and 21, if an enrichment process of a raw feed, being a natural mixture of regular and irregular particles, would be assumed, then variations in shape characteristics for individual enrichment products would be significantly greater.

These differences could be greater and more visible for a 3D dimensioning technique, which takes into account the particle size distribution and shape of all three dimensions, and not, as in the case of DIA technique, two random results, depending on the orientation of the particle in front of the lens. It seems, therefore, that measurements of the size and shape for irregular (strongly flat) particles should be carried out as far as possible by 3D technique. An application of dynamic image analysis techniques for this purpose, as proven in this paper, enables a reliable indication of differences in the

geometry of these particles and obtaining representative results, provided, however, a large population of registered particles (minimum of a few thousand) is used. In such cases, an application of static video measurement technique and the shape of particles, i.e., microscopic image analysis, would be incorrect, because in practice, it is difficult to ensure the representativeness of the sample, and the registered particles will be placed in the most stable position without the possibility of registering the finest dimension, thereby indicating the flatness (irregularities) of these particles. This issue will be a subject of further investigations.

Differences in the structure and geometry of regular and irregular particles between the upper and lower layers of the jig's bed recorded by means of a video measuring system utilizing the DIA technique, and partially proven through analysis of dust emission, showed that the effect of enrichment of chalcedonite particles occurs even within the same categories of regular and irregular particles.

## 7. Patents

Two patents granted in Poland were utilized in the paper:

(1) Author: Gawenda T. Title: *Układ urządzeń do produkcji kruszyw foremnych,* AGH w Krakowie. Wynalazek zgłoszony pod nr P.408045 z dnia 28.04.2014. Patent nr PL233689 granted 8.07.2019
(2) Authors: Gawenda T., Saramak D., Naziemiec Z. Title: *Układ urządzeń do produkcji kruszyw oraz sposób produkcji kruszyw.* AGH w Krakowie, Zgłoszono w UP 10.11.2016 r. pod nr 419 430, Patent nr PL 233318B1 granted 7. 06. 2019.

**Author Contributions:** Conceptualization: T.G. and D.K.; methodology, D.K.; formal analysis: T.G. and A.N.; investigations: T.G. A.K. and A.N. (jig tests), D.K. (image analysis), A.S. (dust emission); writing—original draft preparation: T.G., D.K. and A.S.; writing—review and editing: A.S.; visualization: D.K. and A.K.; supervision: T.G. All authors have read and agreed to the published version of the manuscript.

**Funding:** The article was written within the frames of national Polish project granted by Polish Research Centre: Artykuł jest wynikiem realizacji projektu w ramach konkursu NCBiR: konkursu nr 1 w ramach Poddziałania 4.1.4 "Projekty aplikacyjne" POIR w 2017 r., pt.: Opracowanie i budowa zestawu prototypowych urządzeń technologicznych do budowy innowacyjnego układu technologicznego do uszlachetniania kruszyw mineralnych wraz z przeprowadzeniem ich testów w warunkach zbliżonych do rzeczywistych". Projekt współfinansowany przez Unię Europejską ze środków Europejskiego Funduszu Rozwoju Regionalnego w ramach Działania 4.1 Programu Operacyjnego Inteligentny Rozwój 2014-2020.

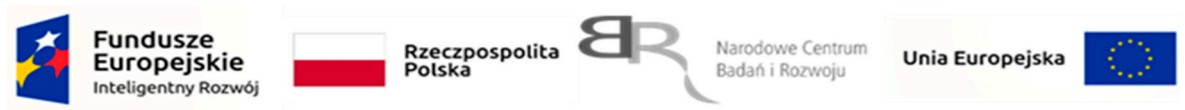

**Conflicts of Interest:** The authors declare no conflict of interest.

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
