# Peer review of "Application of Dynamic Analysis Methods into Assessment of Geometric Properties of Chalcedonite Aggregates Obtained by Means of Gravitational Upgrading Operations"

_minerals, doi:10.3390/min10020180_

Round 1

Reviewer 1 Report

Review Report

Paper Title: Application of dynamic analysis methods into assessment of geometric properties of chalcedonite aggregates obtained by means of gravitational upgrading operations

Ref: minerals-657958

The manuscript has addressed the vision system in the context of their application in the assessment of enrichment operations carried out in a water jig. The geometrical properties of regular and irregular particles of chalcedonite enrichment products carried out in a laboratory ring jig. The authors have investigated the aggregate enrichment, along with a visual analysis made for the obtained products, separately for regular and irregular particles. They have calculated several shape coefficients, particle size distributions in terms of innovative ideas of Feret’s diameter, and determined the intensity of dust emission by individual products.

The authors have concluded that the measurement technique and the particle shape have a significant influence on particle size distribution results considering the vision method. They found repetitive results of the particle size and shape using dynamic image analysis (DIA) that allowed to differentiate the regular and irregular grains. They compared the results with other techniques such as 3D dimensioning technique and static video measurement technique and 2D shape of particles. The authors concluded that the static video and 2D (i.e. microscopic image analysis) shape analysis could not give us reliable results because these cases the particles placed in the most stable position during the measurement.

Overall, the manuscript is impressive, and the authors have made a substantial effort to bring the dissimilarities of using different image measurement techniques. The literature survey is relevant for this study. Methodology and results and subsequent discussions are clear and concise. Therefore, in my opinion, the manuscript can be published in the Journal of Minerals if the authors considered a list of corrections and comments below:

Authors worked on particles with different shapes, where the particles in Figure 2 shows the spheres. It is good if the authors replace the spherical particles by angular/irregular ones. Also, the image quality needs to be improved.

What is the basis of denoting irregular particles as “ZN” and regular particles as “ZF” (refer to line 107)?

What is the threshold criteria for dividing regular and irregular shapes showed in Figures 7 and Figure 8? Was it polygonality, aspect ration or something else? MATLAB program can easily analyze this 2D image and gives AR and polygonality values reliably!

Vision system/method may be replaced by Visual System/Method across the manuscript.

Author Response

Dear Reviewer,

Thank you for your comments on our paper. Please find below the answers. The manuscript was corrected according to your suggestions as follow:

New Figure 2 was created. We hope in such form it possibly meets the quality requirements and correctly reflects the sittuation - particle shape ZN (originally denoting "irregular particles) and ZF (originally denoting "regular particles"), were corrected accordingly: Irr, Reg. The threshold criteria and procedures used in determination whether a particle can be assumed as regular/irregular were given in lines: 134-138 and 175-181. "vision" was replaced by "visual" through the entire manuscript.

All correction in manuscript were introduced with using of red fonts.

On behalf of all Authors,

Kind regards,

Agnieszka Saramak

Reviewer 2 Report

Dynamic shape analysis is now a fairly mature and widely used tool, with much information already available in the literature. The manuscript, as presented, has several flaws that prevent it from being acceptable for publication, even after revision. The motivation for the work is very poor, since the authors do not even explain what they expected from applying jigging to separate the samples. Further, the differences in shape characteristics of the products from different layers are almost absent and are certainly within the experimental error of the technique. No information, for instance, is presented on the jigging process adopted, besides the jigging time. 

I believe the contribution is to knowledge is too small, that the scope is too limited, the hypothesis that based the work was questionable (that jigging can segregate in respect to shape in a way that dynamic analysis can detect) and that the findings are trivial.

Author Response

Dear Reviewer,

Thank you for your comments upon our manuscript. We have improved the paper and added some comments and explanations concerning your suggestions.

An introduction section was supplemented by a comment explaining an idea of application the DIA into the investigation over jigging process products. We totally agree that the technique is mature and well known. However its utilization in assessment of jig product, especially in terms of particle shape, is not common. It can be also a good supplement of mechanical methods controlling the shape of separation products. It was mentioned in lines 72-77 and 180-181.

A more clear description concerning an adopting the jig process, together with a device characteristics, was also added (lines 122-126 and 134-138).

Suitable comments, concerning the problem of significance/insiginificance in shape variations in individual separation products  were  also added (lines 337-341 and 303-305).

We have also added some description into methodology of determination shape regularity/irregularity of product particles (lines 134-138, 175-181).

Kind regards,

Agnieszka Saramak

Round 2

Reviewer 2 Report

The authors properly addressed several of the issues raised. In my opinion, this manuscript gives a fairly limited to knowledge, but at least now it is properly presented and justified.